# Effect of intraoperative Hartmann's versus hypotonic solution administration on FLACC pain scale scores in children: A prospective randomized controlled trial

**Mihyun Kim, Jiyoung Lee, Sungwon Yang, Minsoo Lee, Min Suk Chae[ID], Hyungmook Lee[ID]** *

Department of Anesthesiology and Pain Medicine, Seoul St. Mary's Hospital, College of Medicine, The Catholic University of Korea, Seoul, Republic of Korea

* warmy0828@catholic.ac.kr

## Abstract

### Background

In healthy children, an isotonic solution containing no glucose or a small amount of glucose (1–2%) has been recommended as an intraoperative maintenance fluid due to the potential risk of hyponatremia associated with hypotonic solutions. However, a hypotonic solution with glucose is still widely used as a maintenance fluid for pediatric anesthesia. We speculated that the hypotonic solution may worsen postoperative discomfort and irritability in pediatric patients due to hyponatremia.

### Patients and methods

In the current study, we compared the post-operative Face, Legs, Activity, Cry, Consolability(FLACC) scale scores of pediatric patients aged 3–10 years who received either a 1:2 dextrose solution or Hartmann's solution during Nuss Bar removal.

### Results

The FLACC scale score in the post-anesthesia care unit was higher in the 1:2 dextrose solution group(HYPO) (n = 20) than in the Hartmann's solution group(ISO) (n = 20) (6.30 vs 4.70, p = 0.044, mean difference and 95% Confidence Interval(CI) was 1.6 (0.04 to 3.16)). We also compared opioid consumption at the post-anesthesia care unit. Total dose of fentanyl per body weight in the post-anesthesia care unit was also higher in the HYPO (0.59 vs 0.37 mcg/kg, p = 0.042, mean difference and 95% CI was 0.22 mcg/kg (0.030 to 0.402)).

### Conclusions

Intraoperative use of the hypotonic solution in children causes increased FLACC scale scores, leading to higher opioid consumption in post-anesthesia care unit.

**Data Availability Statement:** All data are available from the Mendeley database. (DOI: 10.17632/63mbdj7dyd.2)

**Funding:** The author(s) received no specific funding for this work.

**Competing interests:** The authors have declared that no competing interests exist.

## Introduction

In healthy children, an isotonic solution containing no glucose or a small amount of glucose (1–2%) has been recommended as an intraoperative maintenance fluid due to the potential risk of hyponatremia associated with hypotonic solutions [1]. A recent systematic review discussed, however, that the existing evidence does not suffice to conclude on the optimal intraoperative fluid for children [2]. Moreover, some current practice guidelines recommend a hypotonic solution as a pediatric intraoperative maintenance fluid [3,4]. As a result, a hypotonic solution with glucose is still widely used as a maintenance fluid for pediatric patients, including in the author's institution [5]. It is based on a traditional recommendation by Holiday-Segar [6]. However, hypotonic solution could decrease plasma sodium level during induction of general anesthesia [7].We speculated that the hypotonic solution may worsen postoperative discomfort and irritability in pediatric patients due to hyponatremia. In this regard, we hypothesized that infusion with the hypotonic solution may increase patient discomfort in the immediate postoperative period.

## Patients and methods

This study was a pragmatic, single-blind, single-center, prospective, randomized controlled trial performed in two parallel groups. It was pragmatic because we used the same infusion protocol for fluids and opioids that we use in current clinical practice. The study protocol was approved on December 16, 2015, by the institutional review board of St. Mary's Hospital, Seoul, Korea (KC16MISI0682), and was registered with the Clinical Research Information Service (http://cris.nih.go.kr, KCT0002199). The authors confirm that all ongoing and related trials for this intervention are registered. Registering of the current study was delayed after patient enrollment of participants started because of miscommunication between authors. Written informed consent was obtained from the parents or legal guardians of the participants.

We collected data from pediatric patients who underwent a Nuss bar removal operation between February 2016 and December 2016. Patient recruitment was done between February 2016 and October 2016. Follow-up was not necessary for the current study. Inclusion criteria were age between 3 and 10 years, American Society of Anesthesiologists physical status I, and only one Nuss bar in the chest. Patients with any other comorbidities and preoperative laboratory abnormalities were excluded. We chose Nuss bar removal as the study operation because only one experienced surgeon performs it at our institution and the procedure is highly standardized, meaning that operation times, intraoperative blood loss, and the dose of anesthetic agents and infused fluid were relatively constant with minimal variability among patients.

We compared the effects of an intraoperative 1:2 dextrose solution (hypotonic dextrous solution) with Hartmann's solution(isotonic solution) on the postoperative FLACC scale scores and the change in plasma sodium and glucose immediately prior to and following surgery.

Plasma sodium and glucose levels were assessed the day before surgery (T0; preoperative baseline values), immediately prior to the surgery (T1), and at the end of surgery before awakening the patient (T2).

As a primary outcome, we assessed the FLACC scale to evaluate irritability in the post-anesthesia care unit(PACU). The FLACC scale is simple, well-validated, and minimally affected by interpersonal variation [8]. Nurses measured the FLACC scale score at 30 min after the patient arrived at the PACU. In the PACU, 0.5 mcg/kg of fentanyl was given when the FLACC scale score was more than three. If the patient's irritability was not ameliorated (FLACC scale score ≥ 3) after 5 min, 0.25 mcg/kg of additional fentanyl was given.

The patients were blinded, but the practitioner and observer were not. The patients were randomly assigned to either a HYPO or an ISO in a 1:1 ratio by random numbers generated with "https://www.graphpad.com/quickcalcs/randomize2/". Group allocation was coded and concealed in opaque sealed envelopes until the statistical analyses were completed. The blinding and allocations were done by another anesthesiologist who did not participate in the trial.

All patients arrived at the preoperative waiting room under the administration of hypotonic dextrous solution. In the preoperative waiting room, the fluid was changed to isotonic solution if the patient was assigned to the ISO. Compositions of the hypotonic dextrous solution and isotonic solution are explained in Table 1.

As blood loss is negligible during Nuss bar removal in the author's institution, intraoperative fluid is given as 4 mL/kg/h for bodyweight up to 10 kg, 2 mL/kg/h for 10 to 20 kg body weight, 1 mL/kg/h for body weight over 20 kg. For the first hour of surgery, the supplement dose is the total maintenance dose of the first half of the fasting period. For the rest of the surgery, the supplement dose is one-quarter of the entire fasting period (8 h) [9]. The patients were sedated with 1.5 mg/kg ketamine i.v. in the preoperative waiting room. After they were transferred to the operating room, 1 μg/kg fentanyl and 0.6 mg/kg rocuronium were administered before endotracheal intubation. Anesthesia was maintained with one minimum alveolar concentration of sevoflurane.

To minimize risks to the patient, we set the plasma sodium level limit between 125 mEq/l and 150 mEq/l, and the plasma glucose level limit between 80 mg/dL and 300 mg/dL. If any value measured at the preoperative period or after anesthetic induction exceeded these limits, the patient was to be excluded from the study and appropriate measures were to be undertaken to restore the patient's state.

The primary outcome of this study was the FLACC scale score. The FLACC scale score of 50 pediatric patients in the PACU after Nuss bar operation in our institution was $6.7 \pm 1.8$. A sample size of 20 per group was required to detect a 25% difference in FLACC scale score between the two groups with a 2-sided test using $\alpha = 0.05$ and $\beta = 0.2$, allowing for a 10% drop-out rate.

GraphPad Prism version 7.03 for Windows (GraphPad Software, La Jolla, CA, www.graphpad.com) was used for statistical analyses. Data are given as mean ± standard deviation, or median (range) (for non-normal distributions). A p-value $< 0.05$ was considered to be significant. All reported p-values are based on two-sided tests. For plasma sodium and glucose level at T0, T1, and T2, we performed a repeated measures 2-way ANOVA to determine differences between the groups and over time. For other values, between-group differences were assessed using Student's t-test or Mann-Whitney U test, when appropriate. The correlations between the FLACC scale score and sodium level at baseline and after the anesthetic induction were analyzed using Pearson's correlation coefficient.

**Table 1. Composition of the study fluids.**

|  | Isotonic solution | Hypotonic solution |
|---|---|---|
| Sodium; mEq/l | 130 | 51.3 |
| Chloride; mEq/l | 109 | 51.3 |
| Glucose; g/l | 0 | 33.3 |
| Potassium; mEq/l | 4 | 0 |
| Calcium; mEq/l | 2.7 | 0 |
| Lactate; mEq/l | 28 | 0 |

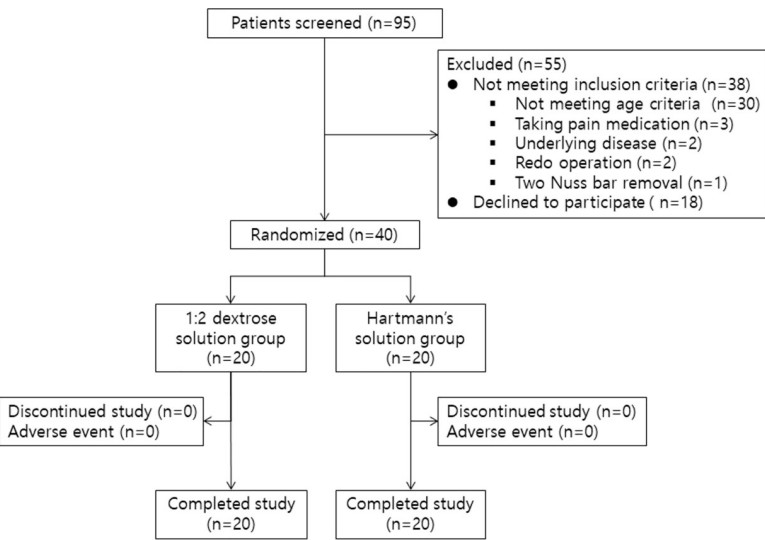

**Fig 1. Consort flow chart.**

## Results

Of the 95 patients who underwent a Nuss bar removal during the study period, 55 were excluded (not meeting inclusion criteria, n = 38; declined to participate, n = 17). Forty patients were finally enrolled and analyzed (Fig 1). Patients' characteristics are shown in Table 2.

There were no adverse events, and all 40 patients enrolled in the study completed the procedures. Total anesthesia time (p = 0.84) and infused fluid per bodyweight (p = 0.76) were comparable between the groups (Table 2). The minimum and maximum sodium values were 132 and 143 mEq/l, and the minimum and maximum glucose values were 86 and 204 mg/dL, respectively.

There were statistically significant differences in the FLACC scale scores in the PACU between the two groups (Fig 2). The mean score was 1.34 times higher in the HYPO. Mean ± SD was 6.30 ± 2.39 in the HYPO and 4.70 ± 2.47 in the ISO (p = 0.044). Mean difference and 95% Confidence Interval(CI) was 1.6 (0.04 to 3.16). There was a weak negative correlation between the FLACC scale score and sodium level after the induction of anaesthesia (r = -0.32, p = 0.043) and at the end of surgery (r = -0.35, p = 0.025). In the HYPO, the correlation between the FLACC scale score and sodium level was higher after anesthesia induction (r = -0.4798, p = 0.032).

The total dose of fentanyl given at PACU was also statistically higher in the HYPO (Fig 3). Fentanyl was given to 17 of 20 patients in the HYPO and 13 of 20 patients in the ISO. The

**Table 2. Baseline characteristics of patients.**

|  | HYPO (n = 20) | ISO (n = 20) |
|---|---|---|
| Age; years | 6(4–10) | 6(5–10) |
| Height; cm | 117.9(104.0–136.5) | 117.6(103.1–143.0) |
| Weight; kg | 21.7(16.2–36.9) | 23.3(16.7–35.1) |
| Sex; male | 15 | 16 |
| Total anesthesia time; minutes | 72.5(38–110) | 75(40–95) |
| Infused fluid per bodyweight; ml/kg | 6.1(3.3–9.1) | 6.3(3.4–7.9) |

Values are median(Range)

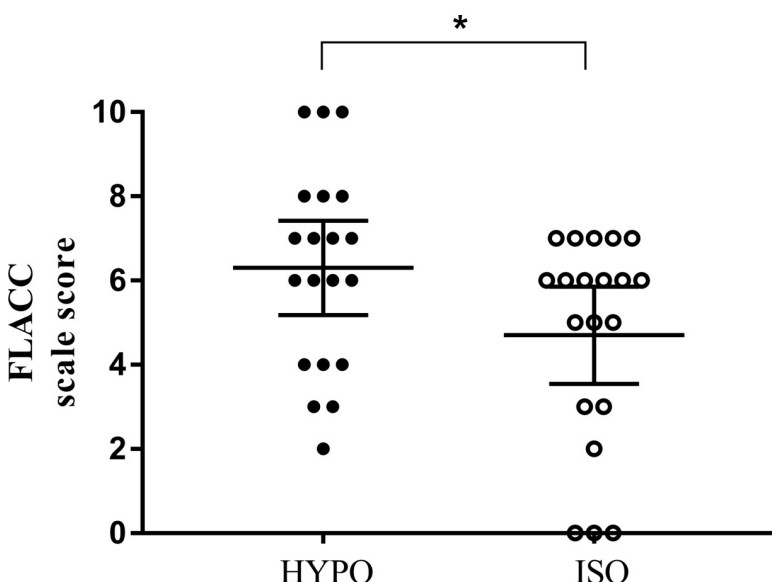

**Fig 2. Face, Legs, Activity, Cry and Consolability(FLACC) scale score in PACU.** Asterisk(*) represents statistically significant values between the two groups.

mean dose of fentanyl was 1.58 times higher in the HYPO. Mean ± SD was 0.59 ± 0.31 in the HYPO and 0.37 ± 0.26 in the ISO (p = 0.042). Mean difference and 95% CI was 0.22mcg/kg (0.030 to 0.402).

Mean plasma sodium level decreased after the induction of anesthesia (T1) in both groups, and it had not recovered by the end of surgery (T2) (Fig 4). In the HYPO, plasma sodium level decreased by 6.4 mEq/l at T1 and by 6.0 mEq/l at T2. In the ISO, plasma sodium level decreased by 3.45 mEq/l at T1 and 2.85 mEq/l at T2. Mean plasma sodium level was significantly lower in the HYPO at both T1 and T2 (Table 3).

Mean plasma glucose level increased significantly in the HYPO after anesthesia induction (Fig 5). Plasma glucose level in the HYPO increased from 108.5 mg/dL to 149.5 mg/dL at T1.

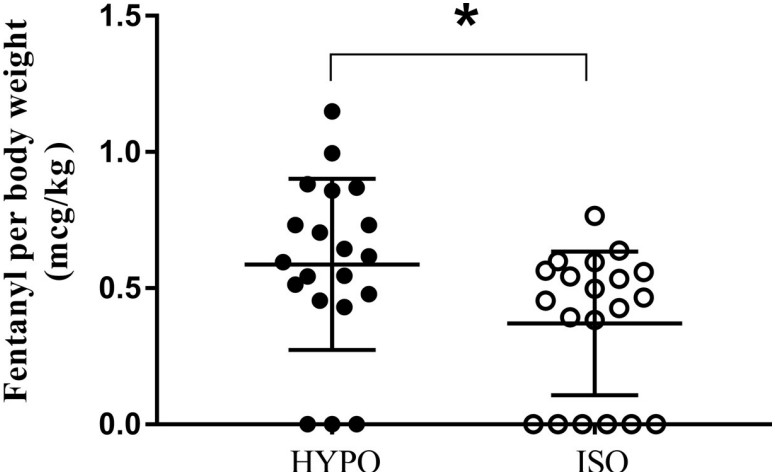

**Fig 3. Fentanyl per body weight at PACU.** Asterisk(*) represents statistically significant values between the two groups.

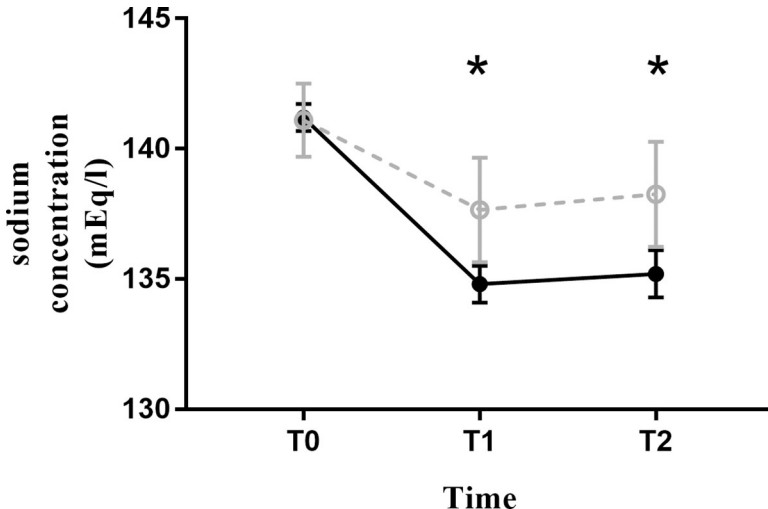

**Fig 4. Plasma sodium concentration.** Solid line: HYPO. Dashed line: ISO. Asterisk(*) represents statistically significant values between the two groups.

It slightly decreased at T2 but remained significantly higher than that at T0 (143.4 mg/dL vs 108.5 mg/dL). In the ISO, mean plasma glucose levels were not changed significantly after anesthesia induction or at the end of surgery. Mean plasma glucose level was significantly higher in the HYPO at both T1 and T2(Table 3).

## Discussion

Acute hyponatremia should be avoided in children because they are more vulnerable to brain swelling due to an increased brain-to-skull size ratio and smaller intracerebral volume of cerebrospinal fluid than adults, which can lead to headache, nausea, vomiting, confusion, and even death [10]. Hypotonic solutions are still prevalently used as a pediatric maintenance fluid [1,5,10–12], however, despite the fact that it can potentially induce hyponatremia.

In the present study, patients treated with a hypotonic solution showed an increased postoperative FLACC scale score compared to those treated with the Hartmann solution. Furthermore, fentanyl use was also higher in the HYPO, which highlights the clinical significance of our results. This is the first published study, to our knowledge, to have examined this issue. In this regard, our results add to the disadvantages of the intraoperative use of hypotonic solutions.

The suspected mechanism underlying the present results is that hyponatremia induced by intraoperative hypotonic solution administration may have caused patient discomfort and irritability, leading to increased FLACC scale scores, because hyponatremia may induce

**Table 3. Mean and SD of plasma sodium and glucose level at T0, T1, and T2.**

|  | HYPO | ISO | Mean difference | CI | P-value |
|---|---|---|---|---|---|
| Sodium_T0 | 141.2 ± 1.11 | 141.1 ± 1.41 | 0.10 | -1.20 to 1.40 | 0.99 |
| Sodium_T1 | 134.8±1.51 | 137.7 ± 2.11 | -2.90 | -4.15 to -1.55 | <0.0001 (*) |
| Sodium_T2 | 135.2 ± 1.94 | 138.2 ± 1.91 | -3.00 | -4.35 to -1.75 | <0.0001 (*) |
| Glucose_T0 | 108.5 ± 15.33 | 101.4 ± 11.55 | 7.15 | -11.27 to 25.27 | 0.73 |
| Glucose_T1 | 149.5 ± 32.23 | 109.0 ± 19.67 | 40.55 | 22.28 to 58.82 | <0.0001 (*) |
| Glucose_T2 | 143.4 ± 38.6 | 102.1 ± 11.3 | 41.30 | 23.03 to 59.57 | <0.0001 (*) |

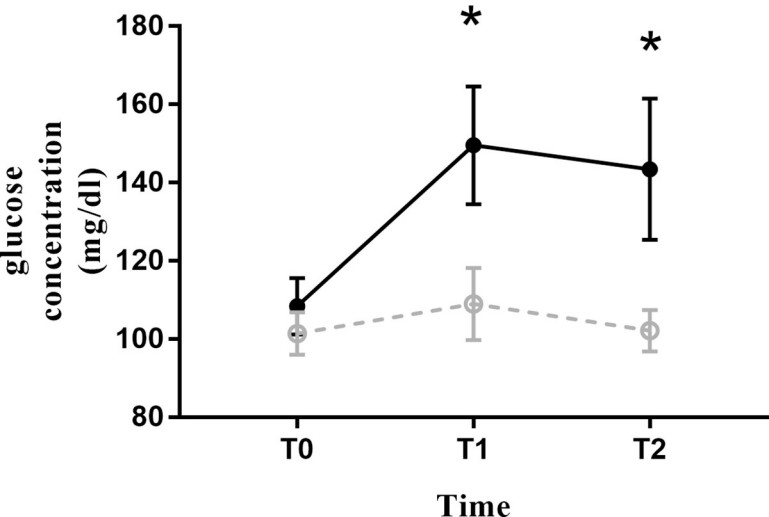

**Fig 5. Plasma glucose concentration.** Solid line: HYPO. Dashed line: ISO. Asterisk(*) represents statistically significant values between the two groups.

irritability and various types of discomfort including headache, chest/abdominal pain, and restlessness [13]. As the basis for this assumption, there was a weak correlation between plasma sodium level at T1 and FLACC scale score in the PACU. Although the FLACC scale is designed for evaluating pain [14], facial grimacing, uneasiness or leg kicking, increased activity, crying, and difficulty consoling the patient (which are the five components of the FLACC scale scoring system) that increase the FLACC score can also be caused by patient discomfort and irritability as a manifestation of hyponatremia.

In the current study, even the short-term use of hypotonic dextrous solution infused during surgery was associated with increased plasma glucose levels and decreased plasma sodium levels. Plasma sodium was abruptly reduced during the induction of anesthesia; which was in line with the findings of a recent observational study [7]. The lowest sodium value in the ISO was 135 mEq/l, suggesting that Hartmann's solution is safer than 1:2 dextrose saline with regard to hyponatremia.

One to two dextrose saline contains 3.3% glucose. While the infusion of fluid without glucose during surgery prevents hyperglycemia [15], there is a risk of hypoglycemia and ketoacidosis. On the other hand, hyperglycemia can increase hypoxic cerebral injuries in pediatric patients and a 5% glucose solution is associated with severe hyperglycemia [16]; thus, solutions containing smaller amounts of glucose are highly preferable [17]. In the current study, the hypotonic dextrous solution increased glucose levels by approximately 37% after anesthesia induction, but they remained relatively constant in the ISO. While the HYPO did exhibit a much higher mean glucose level than the ISO, it was still within the acceptable range. Additionally, both groups exhibited slightly reduced mean glucose levels at the end of surgery.

If we had used 0.9% saline as a maintenance fluid for the control group, the difference of FLACC scale scores could have been larger in degree. However, 0.9% saline is associated with hyperchloremic metabolic acidosis and adverse postoperative outcomes, such as acute kidney injuries [18].

The current study had some limitations. First, it was single-, not double-, blinded (the practitioner and observer were not blinded). However, we did take steps to minimize bias. We used the same volume of fluid and opioid management protocol that we routinely use in our institution and the nurses who obtained the FLACC scale scores were not aware that the

children were involved in a clinical trial. Second, we used ketamine to prevent patient anxiety in the preoperative waiting room, which may also have influenced agitation and pain at the PACU. The dose of ketamine, however, was the same between the groups, resulting in an equal contribution of its effect on both groups. Third, we did not measure plasma sodium levels in the PACU, which may have further elucidated the relationship between plasma sodium levels and the FLACC score.

## Conclusions

Intraoperative use of hypotonic dextrous solution for children caused an increased FLACC pain score, leading to higher opioid use in the acute postoperative period. After confirming this result, we changed our standard clinical procedure. We now use Hartmann's solution instead of hypotonic dextrose solution as the standard intraoperative maintenance fluid in children aged > 1 year.

## Supporting information

**S1 Checklist. CONSORT 2010 checklist of information to include when reporting a randomised trial**[*].
(DOC)

**S1 File.**
(DOC)

**S2 File.**
(DOC)

## Acknowledgments

We would like to thank Editage (www.editage.co.kr) for English language editing.

## Author Contributions

**Conceptualization:** Mihyun Kim, Hyungmook Lee.

**Data curation:** Mihyun Kim, Minsoo Lee, Min Suk Chae.

**Formal analysis:** Jiyoung Lee, Sungwon Yang.

**Investigation:** Mihyun Kim, Jiyoung Lee, Sungwon Yang, Minsoo Lee.

**Methodology:** Mihyun Kim, Hyungmook Lee.

**Project administration:** Hyungmook Lee.

**Resources:** Sungwon Yang, Min Suk Chae.

**Software:** Hyungmook Lee.

**Supervision:** Min Suk Chae.

**Validation:** Mihyun Kim, Jiyoung Lee, Min Suk Chae.

**Visualization:** Sungwon Yang, Minsoo Lee.

**Writing – original draft:** Mihyun Kim.

**Writing – review & editing:** Hyungmook Lee.

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
