## [Decision Letter · Decision Letter 0]

31 Dec 2019

PONE-D-19-19250

Effect of intraoperative Hartmann’s versus hypotonic solution administration on FLACC pain scale scores in children: a prospective randomized controlled trial

PLOS ONE

Dear Dr. Lee,

Thank you for submitting your manuscript to PLOS ONE. After careful consideration, we feel that it has merit but does not fully meet PLOS ONE’s publication criteria as it currently stands. Therefore, we invite you to submit a revised version of the manuscript that addresses the points raised during the review process.

We would appreciate receiving your revised manuscript by Feb 14 2020 11:59PM. To enhance the reproducibility of your results, we recommend that if applicable you deposit your laboratory protocols in protocols.io, where a protocol can be assigned its own identifier (DOI) such that it can be cited independently in the future. For instructions see: http://journals.plos.org/plosone/s/submission-guidelines#loc-laboratory-protocols

We look forward to receiving your revised manuscript.

Kind regards,

Antonio Palazón-Bru, PhD

Academic Editor

PLOS ONE

2. Thank you for submitting your clinical trial to PLOS ONE and for providing the name of the registry and the registration number. The information in the registry entry suggests that your trial was registered after patient recruitment began. PLOS ONE strongly encourages authors to register all trials before recruiting the first participant in a study.

a) your reasons for your delay in registering this study (after enrolment of participants started);

b) confirmation that all related trials are registered by stating: “The authors confirm that all ongoing and related trials for this drug/intervention are registered”.

Please also ensure you report the date at which the ethics committee approved the study as well as the complete date range for patient recruitment and follow-up in the Methods section of your manuscript.

Reviewers' comments:

Reviewer's Responses to Questions

**Comments to the Author**

1. Is the manuscript technically sound, and do the data support the conclusions?

Reviewer #1: Yes

Reviewer #2: Yes

Reviewer #3: Yes

Reviewer #4: Yes

2. Has the statistical analysis been performed appropriately and rigorously? 

Reviewer #1: No

Reviewer #2: I Don't Know

Reviewer #3: Yes

Reviewer #4: Yes

3. Have the authors made all data underlying the findings in their manuscript fully available?

Reviewer #1: Yes

Reviewer #2: No

Reviewer #3: Yes

Reviewer #4: Yes

4. Is the manuscript presented in an intelligible fashion and written in standard English?

Reviewer #1: Yes

Reviewer #2: Yes

Reviewer #3: Yes

Reviewer #4: No

5. Review Comments to the Author

Reviewer #1: Requires a major restructuring

This paper appears to address an important clinical question which has been carefully investigated but cannot be accepted in its current form. The main research question addresses the difference in FLACC score between the two solution groups but this is not given sufficient prominence in the main Results section. Instead, this comparison is somewhat lost among the complex description of less important items. In the Abstract however, it is correctly emphasised as the key comparison.

Points to consider

Line

General It would simplify the presentation if ‘Dextrose solution group’ and ‘Hartmann’s solution’ could be given acronyms, say DEXT and HART, which would then simplify the subsequent text.

32 Move both (n = 20) to Results section of the Abstract

34 Very important that the difference in FLACC mean scores between the groups is quoted together with the 95% CI

34 Usual to quote p-values to two significant figures only – here 0.044

36 Very important that the difference in fentanyl dose mean scores between the groups is quoted together with the 95% CI

36 Usual to quote p-values to two significant figures only – here 0.042

Table 1 Although it is not wrong here to quote mean and standard deviation, here it is usually more informative for the reader to quote mean and range here.

134 -160 Much of this would be better presented in tabular form. As it is, it makes very confusing reading.

General I have counted 18 statistical tests reported in this paper. That is one for every two patients recruited. This is far too many, testing should be confined to the main outcome measures FLACC and Fentanyl and possibly one or two more. The rest of the comparisons should be only made descriptively.

Figs 4 & 5 What is more important than the above comment is that these Figures and their associated text in the Results section should be given more prominence and moved to near the beginning of the results section. To me this is the major flaw in this otherwise good paper.

Reviewer #2: A recommendation has been published to use hypotonic solutions during operations,

rather than isotonic solutions. The authors have conducted a randomized clinical trial in

40 patients in which Group 1 was assigned to a solution that was 1:2 dextrose (hypotonic) versus

Group 2, Hartmann's solution which is 1:2 dextrose [what these ratios actually mean is not

explained.] (isotonic) The published recommendation states that isotonic solutions are

associated with hyponatremia. The primary endpoint is the FLACC scale, which reflects

levels of observed discomfort in the patient. Larger values on the scale indicate more

discomfort. The main conclusion of the paper is that the isotonic (Hartmann's) solution

is better.

The authors appear to stress that the main difference between the solutions is the dextrose

ratios. But are there other constituents of the solutions (NaCl, calcium, etc) which might

account for the observed difference. This may need to be included in the discussion.

The authors should state and describe more clearly which solutions are isotonic and

which are hypotonic, and label them as such in tables and graphs.

There are multiple other outcomes of interest. However there is no indication in the

stat methods section that there was (or wasn't) an adjustment for multiple comparisons.

Whichever it was should be stated clearly.

Reviewer #3: This is an interesting study.

Please refer to the following suggestion to make a good article.

Please list the constitution (sodium, potassium, glucose concentration, and etc) of the two infusions in new table.

Reviewer #4: This is an interesting manuscript where the authors demonstrate that a change from hypotonic to a near isotonic perioperative solution decreases hyponatremia and pain. There are areas of the manuscript that are unclear or could be better explained. Below are my comments.

In both the abstract and the manuscript, the rationale for why 1 – 2% dextrose solution was being used is poorly explained. There really is no rational for it. It is sufficient to say that it has been common practice to administer hypotonic fluids to children intraoperatively based on previous recommendation by Holiday-Segar, or something like that. It is really based on tradition and no sound physiology. Reference 5 is not listed and I doubt it exists. Reference 6 is an outlier and is not reflective of the majority of studies.

In both the abstract and the methods, the design of the study is not explicitly stated. It should explicitly state to compare an intraoperative hypotonic solution with dextrose to Hartmann’s solution to evaluate the FLACC scale score and the change in serum sodium and glucose immediately prior to and following surgery. Or something like that.

Please explain clearly what 1:2 dextrose solution is. What is the dextrose concentration of this fluid and what is the sodium and potassium concentration of this fluid.

The TO, T1 an T2 time points along with what is being measured needs to occur earlier in the methods and be more explicit.

It is not clear what the term waiting room mean. Better to say the preoperative waiting room and recovery room.

The limits of 125 – 150 are too broad. There were probably no patient with a serum sodium of between 125 – 130. The were no patients with a serum sodium < 130 with the lowest sodium in the group 132. Probably better to change this to 130 – 150. Everyone may have had a normal serum sodium at T0, so you could also say 135 -145 if it was true.

The statement about desalinization does not make any sense and should be removed. Saline is no more likely to call desalinization then a more hypotonic fluids and is not a reason not to use it.

6. PLOS authors have the option to publish the peer review history of their article (what does this mean?). If published, this will include your full peer review and any attached files.

Reviewer #1: Yes: David Machin

Reviewer #2: No

Reviewer #3: Yes: Kentaro Ouchi

Reviewer #4: No

---

## [Author Response · Author response to Decision Letter 0]

11 Feb 2020

Response to Reviewers

To Editor : 

a) your reasons for your delay in registering this study (after enrolment of participants started);

:It was because of miscommunication between authors. We thought the trial was registered but it was not. After we found it, we registered the study at once. ( line 65-66 )

b) confirmation that all related trials are registered by stating: “The authors confirm that all ongoing and related trials for this drug/intervention are registered”.

:We added the suggested sentence. ( line 64-65 )

Please also ensure you report the date at which the ethics committee approved the study as well as the complete date range for patient recruitment and follow-up in the Methods section of your manuscript.

:We added an arrival date by the ethics committee. ( line 62 )

: We added the complete date range for patient recruitment. Follow-up of the patient was not necessary because the study was done only in the operation room and the post anesthesia care unit. We also mentioned it. ( line 69-70)

: We deposited the data to public repository. (DOI : http://dx.doi.org/10.17632/63mbdj7dyd.2 )

: To mask the patients’ information, We omitted the operation date and rearranged the order of data.

 

To : Reviewer #1

General It would simplify the presentation if ‘Dextrose solution group’ and ‘Hartmann’s solution’ could be given acronyms, say DEXT and HART, which would then simplify the subsequent text.

: We assign “HYPO” to 1:2 Dextrose solution group and “ISO” to Hartmann’s solution group. We think it is more appropriate to use “HYPO” instead of “DEXT” because the primary interest of the current study is sodium.

32 Move both (n = 20) to Results section of the Abstract

: We modified the manuscript as suggested by Reviewer #1 (line 34)

34 Very important that the difference in FLACC mean scores between the groups is quoted together with the 95% CI

: We add the difference in FLACC mean scores between the groups together with the 95% CI both in abstract and result section as suggested by Reviewer #1 ( line 35, line 146-147 )

34 Usual to quote p-values to two significant figures only – here 0.044

: We modified the manuscript as suggested by Reviewer #1 ( line 35 )

36 Very important that the difference in fentanyl dose mean scores between the groups is quoted together with the 95% CI

: We add the difference in fentanyl dose mean scores between the groups together with the 95% CI as suggested by Reviewer #1 ( line 38, 154 )

36 Usual to quote p-values to two significant figures only – here 0.042

: We modified the manuscript as suggested by Reviewer #1 ( line 37,154 )

Table 1 Although it is not wrong here to quote mean and standard deviation, here it is usually more informative for the reader to quote mean and range here.

: We substitute range for standard deviation as suggested by Reviewer #1. ( line 99 )

: We remove preoperative plasma sodium & glucose value. We moved them into table 3. We add Total anesthesia time and infused fluid per bodyweight into the table.

: We made a correction for note of the table. The values in the table are median, not mean.

134 -160 Much of this would be better presented in tabular form. As it is, it makes very confusing reading.

General I have counted 18 statistical tests reported in this paper. That is one for every two patients recruited. This is far too many, testing should be confined to the main outcome measures FLACC and Fentanyl and possibly one or two more. The rest of the comparisons should be only made descriptively.

: We rewrite most of the results section to follow Reviewer #1’s recommendation. We also made a table 3 for plasma sodium and glucose level. ( line 162-175 )

Figs 4 & 5 What is more important than the above comment is that these Figures and their associated text in the Results section should be given more prominence and moved to near the beginning of the results section. To me this is the major flaw in this otherwise good paper.

: We re-arrange the mentioned paragraph into the beginning of the results section as suggested by Reviewer #1. Fig 4 & 5 are now Fig 2 & 3. ( line 144-160 )

 

To : Reviewer #2

The authors appear to stress that the main difference between the solutions is the dextrose ratios. But are there other constituents of the solutions (NaCl, calcium, etc) which might account for the observed difference. This may need to be included in the discussion.

: The constitution of the two infusions is explained in the new table, Table 1.

: The main difference between the solutions is the concentration of sodium and we discuss the risk of the low sodium concentration solution in Discussion section.

The authors should state and describe more clearly which solutions are isotonic and which are hypotonic, and label them as such in tables and graphs.

: We assign “HYPO” to 1:2 Dextrose solution group and “ISO” to Hartmann’s solution group.

: Also we assign “hypotonic dexterous solution” to 1:2 Dextrous solution and “isotonic solution” to Hartmann’s solution.

: We change or add labels in tables and graphs as recommended by reviewer #2

There are multiple other outcomes of interest. However there is no indication in the stat methods section that there was (or wasn't) an adjustment for multiple comparisons.

Whichever it was should be stated clearly.

: We used repeated measure 2-way ANOVA for multiple comparisons. We re-write the paragraph for statistics and include a sentence as follow : 

: “For plasma sodium and glucose level at T0, T1, and T2, we performed a repeated measures 2-way ANOVA to determine differences between the groups and over time.” ( line 121-123 )

 

To :Reviewer #3

Please list the constitution (sodium, potassium, glucose concentration, and etc) of the two infusions in new table.

: The constitution of the two infusions is explained in the new table, Table 1. ( line 99 )

 

To: Reviewer #4

In both the abstract and the manuscript, the rationale for why 1 – 2% dextrose solution was being used is poorly explained. There really is no rational for it. It is sufficient to say that it has been common practice to administer hypotonic fluids to children intraoperatively based on previous recommendation by Holiday-Segar, or something like that. It is really based on tradition and no sound physiology.

: To follow the recommendation from reviewer #4, we add sentences and references as below.

: “It is based on a traditional recommendation by Holiday-Segar[6]. However, hypotonic solution could decrease plasma sodium level during induction of general anesthesia[7].” ( line 50-52 )

Reference 5 is not listed and I doubt it exists. 

: We fix the citation error. ( line 50 )

Reference 6 is an outlier and is not reflective of the majority of studies.

: We remove the reference 6 and related sentences as recommended by reviewer #4

In both the abstract and the methods, the design of the study is not explicitly stated. It should explicitly state to compare an intraoperative hypotonic solution with dextrose to Hartmann’s solution to evaluate the FLACC scale score and the change in serum sodium and glucose immediately prior to and following surgery. Or something like that.

: We add a sentence as below to follow reviewer #4’s suggestion.

: “We compared the effects of an intraoperative 1:2 dextrose solution (hypotonic dextrous solution) with Hartmann’s solution(isotonic solution) on the postoperative FLACC scale scores and the change in plasma sodium and glucose immediately prior to and following surgery.” ( line 77-79 )

Please explain clearly what 1:2 dextrose solution is. What is the dextrose concentration of this fluid and what is the sodium and potassium concentration of this fluid.

: We made a table for the composition of study fluids at Table 1 in Patients and Methods. ( line 99 )

The TO, T1 an T2 time points along with what is being measured needs to occur earlier in the methods and be more explicit.

: We re-write a related sentence and move it more front : ( line 80-82 )

: “Plasma sodium and glucose levels were assessed the day before surgery (T0; preoperative baseline values), immediately prior to the surgery (T1), and at the end of surgery before awakening the patient (T2).”

It is not clear what the term waiting room mean. Better to say the preoperative waiting room and recovery room.

: We change them “wating room” to “preoperative waiting room” as suggested by Reviewer #4. ( line 94,95, 106, 226 )

The limits of 125 – 150 are too broad. There were probably no patient with a serum sodium of between 125 – 130. The were no patients with a serum sodium < 130 with the lowest sodium in the group 132. Probably better to change this to 130 – 150. Everyone may have had a normal serum sodium at T0, so you could also say 135 -145 if it was true.

: When we conduct another clinical trial, we will narrow the range following Reviewer #4's suggestion.

The statement about desalinization does not make any sense and should be removed. Saline is no more likely to call desalinization then a more hypotonic fluids and is not a reason not to use it.

: We removed the entire sentence about salinization and the associated reference as suggested by reviewer #4.

---

## [Decision Letter · Decision Letter 1]

26 Feb 2020

PONE-D-19-19250R1

Effect of intraoperative Hartmann’s versus hypotonic solution administration on FLACC pain scale scores in children: a prospective randomized controlled trial

PLOS ONE

Dear Dr. Lee,

Thank you for submitting your manuscript to PLOS ONE. After careful consideration, we feel that it has merit but does not fully meet PLOS ONE’s publication criteria as it currently stands. Therefore, we invite you to submit a revised version of the manuscript that addresses the points raised during the review process.

We would appreciate receiving your revised manuscript by Apr 11 2020 11:59PM. To enhance the reproducibility of your results, we recommend that if applicable you deposit your laboratory protocols in protocols.io, where a protocol can be assigned its own identifier (DOI) such that it can be cited independently in the future. For instructions see: http://journals.plos.org/plosone/s/submission-guidelines#loc-laboratory-protocols

We look forward to receiving your revised manuscript.

Kind regards,

Antonio Palazón-Bru, PhD

Academic Editor

PLOS ONE

Reviewers' comments:

Reviewer's Responses to Questions

**Comments to the Author**

1. If the authors have adequately addressed your comments raised in a previous round of review and you feel that this manuscript is now acceptable for publication, you may indicate that here to bypass the “Comments to the Author” section, enter your conflict of interest statement in the “Confidential to Editor” section, and submit your "Accept" recommendation.

Reviewer #1: (No Response)

Reviewer #3: All comments have been addressed

Reviewer #4: All comments have been addressed

2. Is the manuscript technically sound, and do the data support the conclusions?

Reviewer #1: Yes

Reviewer #3: Yes

Reviewer #4: Yes

3. Has the statistical analysis been performed appropriately and rigorously? 

Reviewer #1: Yes

Reviewer #3: Yes

Reviewer #4: Yes

4. Have the authors made all data underlying the findings in their manuscript fully available?

Reviewer #1: Yes

Reviewer #3: Yes

Reviewer #4: Yes

5. Is the manuscript presented in an intelligible fashion and written in standard English?

Reviewer #1: (No Response)

Reviewer #3: Yes

Reviewer #4: Yes

6. Review Comments to the Author

Reviewer #1: Table 3 Title should be amended by adding: ‘Mean and SD of plasma …’

Table 3 CI for plasma quote to 2 decimal places only

Table 3 P-values to 2 decimal places: Amend to 0.99 (although strictly 1) and 0.73

Reviewer #3: Thank you for fixing.

It reflects my point.

Accept.

Reviewer #4: The manuscript is much better following the revisions. The aurthors have made the changes requested by the reviewers and I have no further comments.

7. PLOS authors have the option to publish the peer review history of their article (what does this mean?). If published, this will include your full peer review and any attached files.

Reviewer #1: Yes: David Machin

Reviewer #3: Yes: Kentaro Ouchi

Reviewer #4: Yes: Michael L Moritz

---

## [Author Response · Author response to Decision Letter 1]

27 Feb 2020

To reviewer #1:

Table 3 Title should be amended by adding: ‘Mean and SD of plasma …’

Table 3 CI for plasma quote to 2 decimal places only

Table 3 P-values to 2 decimal places: Amend to 0.99 (although strictly 1) and 0.73

-> We mended table 3 as recommended by reviewer #1. ( line 136 in Manuscript )

---

## [Decision Letter · Decision Letter 2]

4 Mar 2020

Effect of intraoperative Hartmann’s versus hypotonic solution administration on FLACC pain scale scores in children: a prospective randomized controlled trial

PONE-D-19-19250R2

Dear Dr. Lee,

We are pleased to inform you that your manuscript has been judged scientifically suitable for publication and will be formally accepted for publication once it complies with all outstanding technical requirements.

With kind regards,

Antonio Palazón-Bru, PhD

Academic Editor

PLOS ONE

Additional Editor Comments (optional):

Reviewers' comments:

Reviewer's Responses to Questions

**Comments to the Author**

1. If the authors have adequately addressed your comments raised in a previous round of review and you feel that this manuscript is now acceptable for publication, you may indicate that here to bypass the “Comments to the Author” section, enter your conflict of interest statement in the “Confidential to Editor” section, and submit your "Accept" recommendation.

Reviewer #1: All comments have been addressed

2. Is the manuscript technically sound, and do the data support the conclusions?

Reviewer #1: (No Response)

3. Has the statistical analysis been performed appropriately and rigorously? 

Reviewer #1: (No Response)

4. Have the authors made all data underlying the findings in their manuscript fully available?

Reviewer #1: (No Response)

5. Is the manuscript presented in an intelligible fashion and written in standard English?

Reviewer #1: (No Response)

6. Review Comments to the Author

Reviewer #1: (No Response)

7. PLOS authors have the option to publish the peer review history of their article (what does this mean?). If published, this will include your full peer review and any attached files.

Reviewer #1: Yes: David Machin

---

## [Editor Report · Acceptance letter]

6 Mar 2020

PONE-D-19-19250R2 

Effect of intraoperative Hartmann’s versus hypotonic solution administration on FLACC pain scale scores in children: a prospective randomized controlled trial 

Dear Dr. Lee:

I am pleased to inform you that your manuscript has been deemed suitable for publication in PLOS ONE. Congratulations! Your manuscript is now with our production department. 

With kind regards,

on behalf of

Dr. Antonio Palazón-Bru 

Academic Editor

PLOS ONE